# Ciranda—An Inclusive Floor Seating Positioning System and Social Enterprise

**DOI:** 10.3390/ijerph17217942

**Published:** 2020-10-29

**Authors:** Tulio Maximo, Erika Foureaux, Xiao Lu Wang, Kenneth N. K. Fong

**Affiliations:** 1School of Design, The Hong Kong Polytechnic University, Hung Hom, Hong Kong, China; 2Instituto Noisinho da Silva, Belo Horizonte 30.411-305, Brazil; erikafoureaux@gmail.com; 3Department of Applied Social Science, The Hong Kong Polytechnic University, Hung Hom, Hong Kong, China; norahxwang@gmail.com; 4Department of Rehabilitation Sciences, The Hong Kong Polytechnic University, Hung Hom, Hong Kong, China; rsnkfong@polyu.edu.hk

**Keywords:** floor seating positioning system, children with disabilities, developmental milestones, co-design, social enterprise, assistive technology

## Abstract

One of the first challenges for many children with physical disabilities is to sit independently. A floor seating positioning system enables this milestone, helping a child to maintain eye level with other children, play and learn on the floor, rectify his or her posture, and, therefore, helps to include the child within his or her social spectrum. Ciranda is the first comprehensive floor seat solution in Brazil to attend to those needs. The project collected anthropometric data from 370 children who were unable to sit without support. A sample of 37 families of these children was visited, observed, and interviewed. A project requirement compiled key insights from the field data to support a multidisciplinary team of collaborators to co-design solutions. The project resulted in two floor seating positioning systems to attend to different needs. One is a social enterprise where the children’s parents and the community build the seat while the child in need and his or her friends engage in entertainment. The other is a salable seat that helps to raise funds for the social enterprise. The model also unravels other challenges common to assistive technologies, such as access to a device and training for the use and maintenance of the device.

## 1. Introduction

Sitting on the floor plays an essential role in childhood development. Children learn to sit without support between four and nine months, standing with assistance between five to eleven months, and standing alone between seven to seventeen months [1]. Sitting, standing, and walking are well-grounded motor developmental milestones that enable the child to explore the surrounding environment [2]. To sit independently is one of the first milestone challenges for many children with disabilities. The causes impeding independent sitting vary with each disability. Common cases happen in children with cerebral palsy, myelomeningocele, and children with multiple disabilities [3,4]. As the floor environment is paramount for early child development and engagement, not being able to sit on the floor often means being left out from playing and learning, and less independence in self-care such as feeding. A solution commonly adopted to allow independent sitting is using a floor seating positioning system.

A floor seating positioning system (FSPS), commonly referred to as a floor seat, corner seat, or corner chair, is considered a type of special sitting furniture [5]. Beyond the aforesaid benefits of sitting, the device helps the child with a disability (CWD) to develop head control, maintain eye level with other children, rectify his or her posture, and in some cases helps the child to use his or her leg muscles in preparation for walking [6]. More importantly, floor seating helps to include the child within his or her social spectrum. Werner [6] (p. 26) discusses issues common to special seating that can often do more harm than good and notes that “the purpose of a special seat should not be to rigidly hold the child in a position that looks ‘good’, but rather to help the child to learn to sit in a position that is beneficial”. He criticizes existing solutions designed to hold children as if they were no more than ‘sacks of potatoes’, tightening the child with various straps. Werner [6] (p. 26) adds that a special sitting “should not be confining but liberating.”

A key aspect of the effective use of positioning systems is to provide training to users and caregivers [7,8,9]. Incorrect use of a positioning system can cause fatigue, discomfort, postural deformities, pressure sores, and even premature death [9,10,11,12]. As a consequence of assistive technology (AT) that is poorly fitted to users’ needs, discontinuance of the device is likely to occur [13,14]. Adapted wheelchairs and FSPSs share similar features, such as the use of an adjustable support system to rectify the posture. Mukherjee and Samanta [13] examined the destiny of 162 donated wheelchairs in India and found that 57.4% of the donated chairs were not used and 14.2% had been sold. Reasons for low usage included fatigue, pain, discomfort, and the unsuitability of the wheelchairs to the climate.

The past decade in Brazil witnessed many advancements with regards to policies towards the inclusion of people with disabilities (PWDs). Two significant forms of progress were the ratification of the United Nation Convention on the Rights of Persons with Disabilities (UN CRPD) [15] and the establishment of the national plan for the rights of disabled people, Viver Sem Limites [16]. The last articulates policies regarding health care, social inclusion, accessibility, and access to education. The national plan covers a list of assistive technologies eligible for public provision via healthcare services. The list offers 95 devices, adaptations, and substitutions described and priced. However, the list does not offer a range of devices often needed by a child with disabilities, such as augmented communication devices, standers, and FSPSs [17].

Before the Ciranda Project, the families in need of an FSPS in Brazil had to rely on imported salable solutions or bespoke solutions. The price of imported FSPS solutions without shipping and importing costs ranges from 167 USD to several hundreds of dollars depending on the device [18]. Considering that the minimum wage per month in Brazil was around 264 USD during the project development and that more than 60% of Brazilians received below the minimum wage in 2018 [19], importing a device was and still is a distant solution for most Brazilian families. They had to rely on bespoke solutions provided by a few skilled professionals whose services are not widely available. To cover this gap, the Instituto Noisinho da Silva (INS), a non-government organization aiming to include children with disabilities and promote equality by means of design, created the Ciranda Project.

The Ciranda Project’s main objective was to enable as many Brazilian children as possible within ages 0 to 6 years, or children within 110 cm height, to sit independently on the floor. The project developed two solutions with similar functional characteristics to attend to the needs of different user groups. The first is a salable FSPS to benefit the physically disabled children who cannot sit without support and which is affordable for families to purchase. The second is the social enterprise Ciranda Workshop, which was created to benefit physically disabled children who cannot sit without support and whose family cannot afford an assistive device. Social enterprises provide nongovernmental, market-based solutions to social issues or unmet needs [20]. This publication aims to report on the Ciranda Project as a case study of an innovative alternative form of AT design and service provision model.

## 2. Materials and Methods

The project adopted the Research through Design (RtD) approach, a process of iteratively designing artefacts as a creative way of investigating what a potential future might be [21]. RtD integrates knowledge from interdisciplinary teams into a sequence of iteratively planning, acting, observing, and then reflecting. The sequence is often repeated in loops until a best-fit solution is found. RtD also serves to challenge current perceptions on the role and form of technology, advancing the practice of design with the goal of not only creating societal change but improving society at large [21]. INS also adopted inclusive design principles to the project, which is the design of mainstream products and/or services that are accessible to, and usable by, as many people as reasonably possible without the need for special adaptation or specialized design [22]. The project aimed to use the inclusive design mindset to challenge the stigma associated with the use of AT devices, and design solutions that would be financially inclusive and usable by children with different conditions requiring independent seating.

The INS team invited various stakeholders to contribute from early stages until the completion of the project as ‘experts of their experience’. The Ciranda Project team was composed of a few of INS’ in-house designers and engineers, and many volunteer collaborators from a diverse array of professions, including occupational therapists, physiotherapists, social workers, social scientists, plastic molding experts, woodwork technicians, and mechanical engineers. During the collaboratory process, INS’ in-house designers acted as facilitators, informing stakeholders, collecting their insights, instigating their creativity, and summarizing information into project specifications and reports.

The team first conducted field research to collect anthropometric data from the CWDs, interview the CWDs’ families and care institutions, and observe their adopted solutions to sitting. Data was collected in conjunction with INS’s Inclusive School Desk project, for which 3100 children from 28 schools had anthropometric data collected in the metropolitan region of Belo Horizonte, Brazil [23]. Purposive sampling was used to identify CWDs who were unable to sit independently from the Inclusive School Desk project. Apart from the schools, snowball sampling was used to gather more participants by contacting the local prefecture and the local charitable institutions providing care for the CWDs. A total of 370 CWDs that were unable to sit independently had their anthropometric data collected for Ciranda Project.

The INS team then visited 37 families based on the following purposive sampling criteria regarding their children’s characteristics:children with muscle hypotonicity at rest;children with muscle hypertonicity at rest;children with spastic movements;the child with the smallest sit height dimension;the child with the largest sit height dimension;children from low-income families;the child with the largest head circumference;children with asymmetrical body mass distribution.

During the visits, the INS team observed each family’s current seating solutions and interviewed 42 parents and caregivers in order to determine their needs for FSPS and other AT devices, and get an estimation of their purchasing power regarding assistive technologies. The data were analyzed using thematic analyses for uncovering key themes and processed into a list of project requirements. The INS team then developed a series of solutions in the form of sketches, paper and cardboard mock-ups, foam models, and functional prototypes. The INS team transferred crucial measurements from the anthropometric data bank to a cushioned doll, which was used initially to test dimensions and scenarios. The INS team then created a testing group with 12 children accompanied by their parents or caregivers. To select the testing group participants, INS used the same criteria from the field visits, to reflect the diversity of users and characteristics in order to test different aspects of the functional prototypes.

## 3. Results

### 3.1. Key Research Insights

The majority of the families visited were composed of unemployed mothers receiving a social benefit called Continuous Cash Benefit (original term: Benefício de Prestação Continuada) aimed at PWDs. Many of these mothers were single or had little support from their partners, having to rely on other family members and people of their community to support their childcare. Most families used a similar network of mixed service provision to support the child development, composed of public rehabilitation hospitals, not-for-profit rehabilitation hospitals, private clinics supported by government, special education institutions, as well as mainstream public education.

The ATs in use were mostly provided by the government or donated by charitable institutions. Observed families in possession of an adapted wheelchair used it as the main seating base throughout the day, with the parents and caregivers moving the adapted wheelchair to the required location. Many children were still on the government waiting list for an adapted wheelchair, which takes up to 36 months [24,25,26]. Most participants rely on alternative solutions for sitting out of the wheelchair in diversified positions. Adopted solutions for seating on the floor were crafted solutions, such as stuffed jeans (trousers), or a plastic bucked cut into a seat shape. Interview participants often reported leaving the child seated or laid on the sofa as an alternative seating solution. They also reported using cushions and pillows to help to accommodate the child and to change position.

Although observed crafted solutions enable the CWD to sit on the floor, it was not enough to provide a stable and rectified support or adjustability, hence posing a risk to cause more harm than good to spinal growth. Also, crafted solutions did not guarantee the freedom of movement of the upper arms, thus potentially limiting a child’s activities.

### 3.2. The Salable Ciranda

Considering that the access to FSPSs in Brazil was found to be limited and expensive, INS proposed two FSPS solutions: the salable Ciranda seat, and the social enterprise Ciranda Workshop. INS designed the two solutions intermittently. This section will describe how the research insights and the continuous people-centered process informed the design features of the salable Ciranda seat.

The salable Ciranda is equipped with a safety loop design that hugs the child from the chest, stabilizing the trunk to the back seat (Figure 1a). The loop design avoids the negative connotation of tightening the child with various straps [6]. To design a loop that accommodates diverse sizes of children, the INS team consulted the collected anthropometric measures for sitting shoulder height, chest depth, and head breadth. The loop prototypes were tested for any pressure point after 45 m of use and modified until no pressure point was found within the testing group. The safety loop is locked by an internal mechanical clamp activated by an oblong handle located in the back of the seat (Figure 1b). The necessary force to lock the safety loop was decided after testing with mothers and the siblings of CWDs. The required force of application had to be strong enough to avoid another child unlocking the safety loop accidentally, but not so strong that some adults could not lock it.

It was clear that no matter the financial circumstances of the participants, they all make extensive use of the AT devices they possess, commonly using it for different purposes and in different contexts. As a consequence, Ciranda design considered the usage at both internal environments, such as homes and clinics, and external environments, such as parks and beaches. Parents can also use the Ciranda for bathing their child and enjoying the shoreline or other wet areas. To enable these diverse applications, Ciranda utilizes waterproof materials and a set of three removable suction cups attached to the bottom of the seat to provide grip in smooth wet surfaces. Additionally, the INS team designed the seat base to support the addition of legs so that it can be used with the knees bent. However, the leg design has not been finalized and tested, and hence the legs are not yet available on the market.

The device contains two carry handles integrated into the back-seat design in order to facilitate transportation of the assembled seat (Figure 1b). Ciranda can be disassembled for easier transportation (Figure 2b) and carried in a backpack, which is sold separately to reduce the entry costs of the Ciranda seat (Figure 2c). INS invited CWD parents to try assembling and disassembling a functional prototype and a pre-production model. Collected information helped to improve the user manual efficacy and define the necessary force to lock the back seat to the seat. The user-manual is attached to the product to facilitate its access and encourage users to follow safety instructions (Figure 1a). Consulted physiotherapists and occupational therapists recommend a maximum continuous use of 45 min, considering the device is a piece of rectification equipment and can induce pain after prolonged use.

Ciranda has various features to enable the trunk alignment to be closer to the position it would be in if standing. The seat contains a forward angle to induce hip rotation, and two hip supports and one front support, each containing three levels of adjustments. These features reduce extensive posterior pelvic tilt, anterior pelvic tilt, pelvic obliquity, and rotation. To design the back seat, the INS team avoided adopting side wings with a positive angle to keep the body straight, a solution commonly used in similar devices. Consulted occupational therapists and physiotherapists highlighted that side wings could block the child’s side view and interfere in the superior members’ movement as well as in the back alignment. Instead, the back seat has a negative side wing angle to enable upper limbs movement and has a mild lordosis over the lower back.

Various tests and adjustments were made on the pre-production model to ensure that the back seat can flex to alleviate the impact of the head hitting hard against the back seat, in case of involuntary movements. Rotational molding experts’ participation was decisive to enable such a feature.

The first batches of Ciranda were sold for 500 BRL in 2011 (around 297 USD, considering the average quotation in 2011). Ciranda’s current price is 800 BRL (around 203 USD, considering the average quotation in 2019), as a result of the raise in various services and the Brazilian economic instability over the past five years [27]. Although there were advancements in 2015 to reduce taxes from imported AT devices to Brazil [28], the shipping costs and fluctuation of the Brazilian currency puts the salable Ciranda within the most affordable options for FSPSs. A search in the Disabled Living Foundation databank resulted in 10 FSPSs with similar functions to the salable Ciranda, with the costs ranging from 167 USD to 750 USD, with the average price of 430 USD (considering the quotation on 20 October 2020) [18].

### 3.3. The Social Enterprise

The Ciranda Workshop is a social enterprise to cater to the needs of Brazilian families in socio-economical vulnerable conditions and who have a CWD that cannot sit independently. The social enterprise goes beyond donating an AT device and aims to engage the parents and the local community to build their FSPS. As one CWD mother participant of the Ciranda Workshop reported:
Learn[ing] to do it makes all the difference because the satisfaction is much higher. You can look (and think) ‘I did this myself’ and (realize) you have the capacity to do something useful to your child and also to help other mothers... If it is something donated, I [would] have no clue about the work someone put [into] it.

INS initially aimed to produce only a commercial version of FSPS that would be affordable to most Brazilians. However, the initial production budget estimation revealed that although the commercial version would cost less than importing a similar solution, it would still be unaffordable for the majority of Brazilians earning a minimum wage or for families relying on social benefits. As an alternative, INS designed a social enterprise that allows families in vulnerable socio-economic conditions to have access to an FSPS not offered by the government or a local charitable institution. At the same time, the social enterprise can offer training, entertainment, and awareness to the communities surrounding CWDs. The idea was to promote workshops to gather the families and communities of CWDs to build their own FSPS, providing them training to do the following:position the child into the chair;understand the risks and consequences of bad positioning;understands the risks and consequences of the FSPS misuse;care for, maintain, and replicate the Ciranda seat;encourage follow up within participants’ healthcare network.

The Ciranda Workshop seat (Figure 3) uses the same dimension parameters from the salable Ciranda. The main difference is that the Ciranda Workshop seat cannot be used in wet areas, and the base does not contain a forward angle. These features were sacrificed to facilitate the seat production within a workshop event. Another difference aimed at workshop feasibility is that the seat’s loop is divided into two parts, each covered with a washable cushioning. Besides providing comfort, the cushions help to prevent the humidity from upper body parts from being transferred to the loop’s raw material, thereby helping to conserve the loop’s moving parts. Each loop is locked using a metal locking pin attached to the seat handle. The Ciranda Workshop loops have three levels of adjustments personalized to each child, while the salable Ciranda loop can adjust in 180°.

### 3.4. The Ciranda Workshop Event Model

The Ciranda Workshop is a two-day itinerant event organized during weekends through different regions of Brazil. Four stages precede the event, which are as follows:locating the children in need;finding the appropriate materials to build the device;locating workshops in the community;finding local volunteers and partner institutions and hiring additional staff.

The first step is to locate the children in need in the region receiving the event. INS staff first contact the local prefecture and the local charitable institutions providing care for the CWDs to screen potential participants. With the help of these institutions, INS staff screen the participants that may benefit from the device. If the number of children located in the hosting region is below the ideal number, then surrounding regions are contacted until a minimum of 20 families of CWDs are confirmed.

Another critical aspect of a social enterprise of this type is to make use of local and inexpensive materials. In Brazil, the middle-density-fiberboard (MDF) is found in most places, and most workshops can work with the material. MDF is affordable, durable for the purpose, and the fact that there is a broad range of suppliers in Brazil facilitates donations or fair sales prices for non-profitable institutions. It is relatively easy to process the MDF using tools that are easy to transport, such as sandpaper, chisels, and white glue. By using MDF, any workshop with benchwork for woodwork can host the event.

The last stage is to find local volunteers and partners to support the event and to hire additional staff to complement the INS itinerant team. The primary type of volunteer required is local caregivers, often recommended by the local charitable institutions. Additionally, occupational therapists, workshop technicians, entertainers, and organizers are welcomed to support. The social enterprise needs to keep a safe budget to hire staff in order to complement the number of volunteers.

Once the preparatory stages have been completed, the event can take place. INS invites the families to bring their child and friends to the event, with transport, food, and entertainment arranged. The event is structured into two parallel sections: a workshop for the parents to build their own Ciranda seat, and entertainment activities for the CWDs and friends, supported by local caregivers and local entertainers. On the first day (see Table 1), while the children join cultural and entertainment activities, the parents receive a take-home kit, including a uniform and a manual with instructions to maintain their Ciranda seat and to replicate the Ciranda for other CWDs in need. Added to the kit are the required tools to build the Ciranda seat, which are returned at the end of the event (Figure 3). INS starts the workshop with an explanation of the safety instructions and overall guidelines. One fully finished Ciranda seat is presented for quality reference (Figure 3). The parents are given one piece at a time to work on, and they can only proceed to the next piece after all the parents have completed the same piece. The aim is to stimulate parents to get to know each other and collaborate with each other. At every new piece, the INS staff provide more instructions regarding the working method. By the end of the first day, most pieces have been completed and are ready to assemble.

On the second day (see Table 2), the parents finish all the parts and assemble the Ciranda seat during the morning (Figure 4) while the children engage in more cultural and entertainment activities. In the afternoon, after a lunch break, each family and child are called to attend a training section. There, the child tests the seat, and the INS staff make individual adjustments, if needed. The INS staff then teach the parents how to position the child into the chair, explain the risks and consequences of wrong positioning and misuse of the equipment, and explain how to maintain the Ciranda. INS also encourages parents to follow up with healthcare professionals to evaluate their rehabilitation goals.

In the first round of Ciranda events, a closing ceremony was organized after all parents received the training to thank the sponsors and collaborators. The ceremony was adapted to each family during the training section to avoid keeping the CWDs beyond the necessary time. In the ceremony, the family are invited for a group picture to receive the certificate of participation, proving that they are ready to reproduce the Ciranda seat for others in need (Table 2).

## 4. Discussion

The Ciranda model shed light on a few constraints typical to AT design and service provision, such as access barriers, lack of commercial value, discontinuance of use, and need for training for the use and maintenance of the device [9,10,11,12,13,14,15].

The adoption of RtD and inclusive strategies proved to help with overcoming some of the industrialization and market barriers common to ATs. The sealable Ciranda was able to reach the intended price and various stakeholders needs, such as accommodating the needs of children of various disabilities as well as attending to the limitations of the rotational molding process, only because of the close cooperation between stakeholders. Assistive technology devices often have a small number of potential users for the same device, leading to small market niches and difficulties in attracting investments [8,29,30,31]. An inclusive design strategy can help businesses to define and implement meaningful ways of engaging with people that will enable a deeper understanding of a particular market sector and what they might demand from their product or service [32]. A limitation of using the inclusive design approach for the salable Ciranda design is that it does not provide full adjustability, for example, for children in need of a trunk side pad to reduce the symptoms of scoliosis. The Ciranda Workshop seat shares a similar limitation. Although the social enterprise allows personal adjustments, these are limited by various factors, such as time for evaluating, testing, and making significant personalized adjustments. Only minor adjustments are possible, for example, inserting a cushioning for the head for children with spastic movements.

Despite the success of Ciranda’s aimed price, INS has been struggling to sustain regular profits for the salable Ciranda as well as to replicate the Ciranda Workshop in recent years. One reason is that Brazilian government’s list of AT subsidized devices does not cover FSPSs such as the Ciranda seat. Hence, it is rare to receive orders in large quantities. Additionally, Brazil’s economy has been undergoing continuous difficulties since 2014 [27], affecting the consumption of the salable Ciranda as well as donations to replicate the Ciranda social enterprise. Hence, one project limitation is financial sustainability, as the profits from the salable Ciranda are not enough to sustain the social enterprise. The often-used approach to “buy one, give one” would make the salable Ciranda unaffordable for many families, increasing the need for the Oficina da Ciranda. There is a need to improve the financial sustainability of Ciranda’s model to rely less on donations.

Andrich et al. [7] discuss the importance of a service delivery system when acquiring an assistive technology due to the financial issue related to the need to remove cost barriers and to allow equal opportunities in the access of those who need them. The shortage of AT options is worsened when devices and services do not fit the list or criteria for AT provision from governments and service providers. Considering that the Brazilian government’s current assistive technology service delivery system does not provide FSPSs, the Ciranda Workshop complements existing services to remove cost barriers and allow equal opportunities to FSPS access in Brazil. The Ciranda Workshop model, where the community is engaged to build their own AT solutions, can alleviate the burden on government institutions, especially in developing countries. Specifically, positioned on the margins of formal government agencies and sometimes even beyond their purview, civil society initiatives such as community enterprises or social enterprises are playing an expanding role in the provision of goods and services in particular communities, often rural places [33,34,35,36,37]. The local circumstance of socially marginalized or geographically remote communities is typically very different from what centralized development decision-makers assume, and there is a lack of market presence [33,34,35,36,37]. Hence, whether the AT device and service provision are designed for-profit or not-for-profit, it must be designed based on evidence and in participation with PWDs and relevant stakeholders. Therefore, it is pivotal that organizations developing ATs or providing AT services cooperate with other institutions, such as universities, governments, and charitable institutions providing care for the PWDs.

Last but not least, the Ciranda Project provides an alternative solution to the traditional AT service delivery system, which is pivotal for training on the use and maintenance of the device [7,8,9] as well as for avoiding misuse or discontinuance due to the device not fitting user needs and expectations [10,11,12,13,14]. This is reflected in the co-design of the salable Ciranda with caregivers, CWDs, and other stakeholders, which has helped with finding suitable solutions to avoid device misuse and discontinuance, which is common with self-purchased ATs. Similarly, the Ciranda Workshop engages the caregivers and the community in the construction of their CWD device, which in turn creates a sense of ownership that counteracts discontinuance. Donation of AT device without appropriate assessment and training for its use can lead to devices that can cause more harm than good [6,9,10,12,13]. The Ciranda Workshop event provides an opportunity to train the parents on the correct use and maintenance of the device. Although the parents are encouraged to follow up the Ciranda’s use with rehabilitation organizations for CWDs, this is not guaranteed, which makes it difficult to track the effectiveness of the device with regards to the CWD rehabilitation goals. Alternative solutions to traditional service delivery systems should account for this point.

## 5. Conclusions

The Ciranda Project model uses strategies from the Research through Design approach and inclusive design to provide an alternative solution to the limitations that are typical to assistive technology design and service provision. Such limitations often include lack of commercial value, barriers to access assistive technology devices and services, appropriate training for the use and maintenance of the device, as well as discontinuance of the use of assistive technology due to the device not fitting user needs and expectations. The project is the result of the close cooperation between various stakeholders and provided two solutions to enable children with disabilities to sit independently. One solution is a social enterprise for families in socio-economical vulnerable conditions where the children’s parents and the community build their equipment and receive training. The other is a salable seat aimed at both indoor and outdoor use, which also helps to raise fund for the social enterprise. The Ciranda Project model can alleviate the burden of healthcare systems, especially in low- and middle-income countries and countries without a national assistive technology policy as well as in countries that do not provide floor seating positioning systems widely.

## 6. Patents

This project resulted in the industrial design patent number PI0703151-3 A2 submitted to the Instituto Nacional de Propriedade Industrial (INPI), Brazil.

## Figures and Tables

**Figure 1 ijerph-17-07942-f001:**
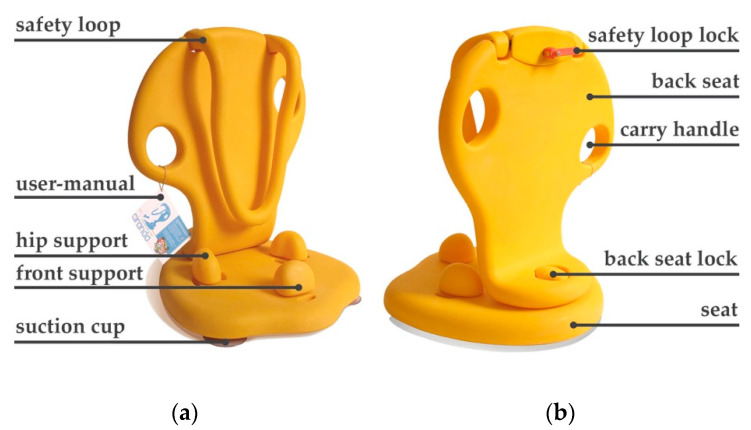
Salable Ciranda parts: (**a**) front of Ciranda seat, indicating the safety loop, user-manual, hip support, front support, and suction cup; (**b**) back of Ciranda seat, indicating the safety loop lock, back seat, carry handle, back seat lock, and seat.

**Figure 2 ijerph-17-07942-f002:**
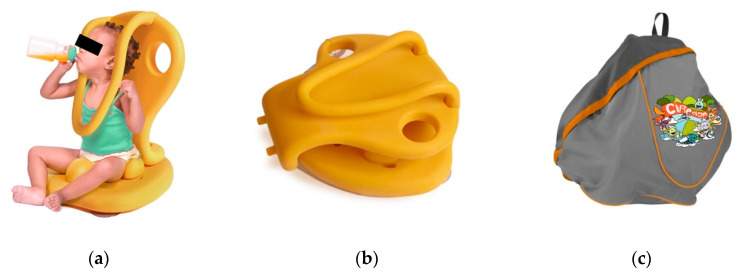
Salable Ciranda: (**a**) Ciranda seat with a child sitting and drinking a bottle independently; (**b**) Ciranda seat with the backrest disassembled and placed over the seat with parts assembled; (**c**) Ciranda seat backpack.

**Figure 3 ijerph-17-07942-f003:**
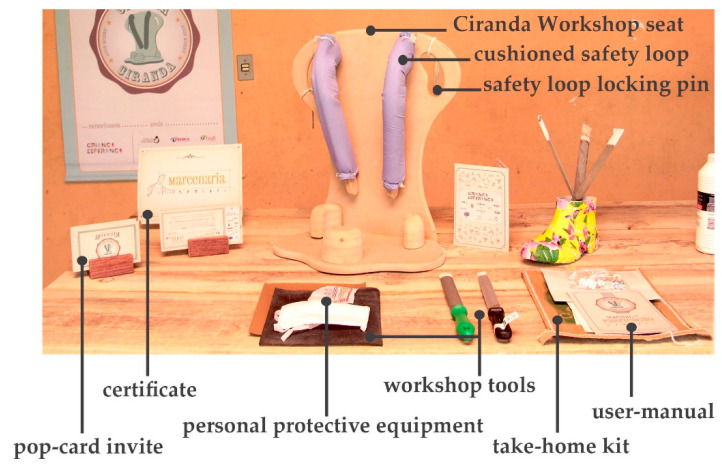
Ciranda Workshop concluded seat and the items used to reproduce the seat.

**Figure 4 ijerph-17-07942-f004:**
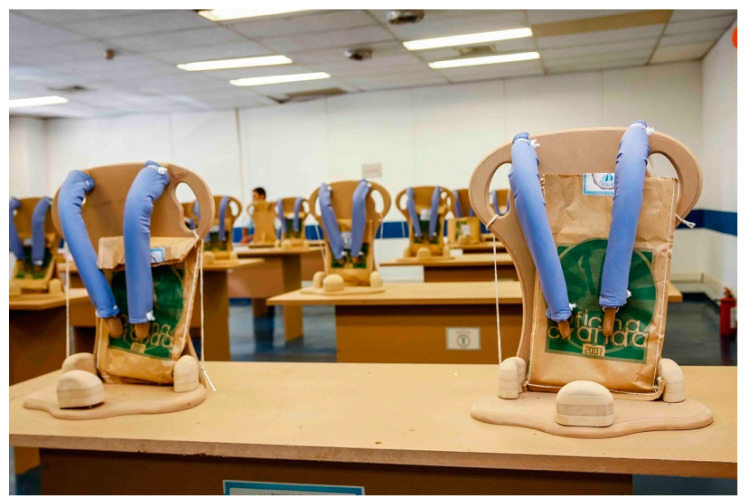
Concluded Ciranda seats during the second day of the Ciranda Workshop.

**Table 1 ijerph-17-07942-t001:** The first day of the Ciranda Workshop event.

Day 1	
Seat Production	Entertainment
** 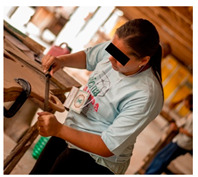 **	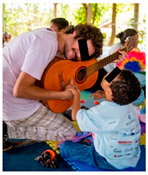
Each family receives a kit and required tools, then receives training for each production step	Entertainers, volunteers, and family members join the CWDs for fun

**Table 2 ijerph-17-07942-t002:** The second day of the Ciranda Workshop event.

Day 2	
**Seat Conclusion**	**Entertainment**
** 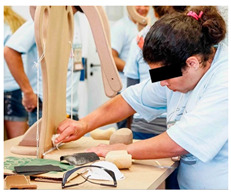 **	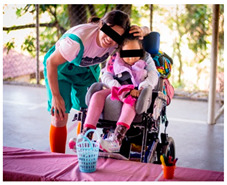
The parents finish and assemble the Ciranda parts in the morning	The CWDs and friends join more cultural and entertainment activities in the morning
**Seat test**	**Training and closing**
** 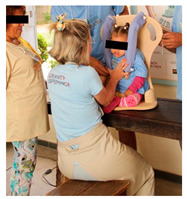 **	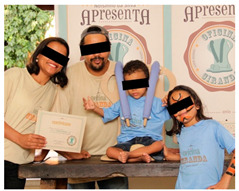
Each CWD tests the Ciranda seat, and individual adjustments are made, as necessary	Parents receive training and a certificate, closing the event

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
