# Peer review of "Ciranda—An Inclusive Floor Seating Positioning System and Social Enterprise"

_ijerph, 2020, doi:10.3390/ijerph17217942_

Round 1
Reviewer 1 Report
Dear authors:
To be part of a group, and for a child to sit on the floor with other children can be part of social interaction. The approach to built the seat in a workshop together with the parents is a great possibility not only for low-income countries. The possibility to built something that their child can use in daily life situation could help the parents too, to support their child with disability
Introduction
Page 1/line 28/29: You could add another citation for the timepoint where children were able to sit up independently, by them own.
Materials and Method:
Page 3/Line 106-113: You mention different groups of children with different disabilities. It would be nice to know if you plan to add specifically adaption for individual solutions. As every child is different, they need maybe slightly different chairs. This would be a point to discuss at the end.
Results
Page 3/line 115: Please add a summary table of the results of the interview, and observation of the 370 children; e.g. needs, issues, actual situation
Page 7/Picture Seat test
Hear the seat is positioned on a table. The legs of the child are in flexion. This has an influence of his posture, and pelvis position. If the chair is used flat on the floor, it should be tested flat on the floor, or on a table but in the same leg position of the child as used later on.
Additional comment
Many children, especially children with spastic or dystonic cerebral palsy, can sit better, and more actively (more independent) upright if the knees are bent (90°). Therefore, a small adaption could be to put the seat on a small box. So, the chair can still be placed on the floor.
Further workshops could be held with the parents to work out how the children can get in and out of the seat and how the children can also actively participate.
Reviewer 2 Report
Tha paper presents an innovative sit for children with physical disabilities.
The approach is interesting but more design data and parameters are needed explaining the reason of the design choice.
More data are needed on the experimental part or social part to allow the reader understand the effectiveness of the proposal.
I think that the proposal has a lot of potential but the mentioned point must be improved to show them to the reader.
Reviewer 3 Report
This work presents a model for floor seat solution to attend children who cannot sit without support (implemented in Brazil). The project resulted in two floor-seating positioning systems, one is a social enterprise where the children parents and the community build their seat, and the other is a salable seat that helps to raise fund for the social enterprise.
The project is relevant for the knowledge area. Nevertheless, some issues should be addressed before the manuscript could be considered for publication:
Figure 2. Please confirm that author have permission to display the child image.
Line 66 Stated that “importing costs ranges from 250 USD”. Line 182 Stated that “Ciranda (around 191USD), is still more affordable than most imported solutions”. Please clarify if some imported solutions are under 191USD.
Integration Costs, for the Ciranda floor seat, should be provided.
Mechanical diagrams, for the Ciranda floor seat components, should be provided.
General guidelines for each production step should be provided.
Round 2
Reviewer 2 Report
Some desiggn parameters are still missing the readear cannot still understand the dimension the weight and why these parameteres have been choosen in such a way.
The paper improved and the authors followed the reviewers' suggestion but the mentioned information are still missing.
Before pubblishing please add the details.
All in all after this it can be accepted.
Reviewer 3 Report
The authors addressed all the review comments. I would recommend considering this manuscript for publication.